# Habitat use of Bechstein´s bats (*Myotis bechsteinii*) within wind parks in forests

**Johanna Hurst**[1]*, **Fränzi Korner-Nievergelt**[2], **Robert Brinkmann**[1]

**1** Freiburg Institute of Applied Animal Ecology, Freiburg, Germany, **2** Landscape Ecology and Ecosystem Conservation, University of Basel, Basel, Switzerland

* hurst@frinat.de

## Abstract

Wind parks are increasingly installed at forest sites, which are a sensitive habitat for bats. As a consequence forest habitats are disturbed and altered by logging, edge effects and turbine operation noises. The Bechstein´s bat (*Myotis bechsteinii*) is highly dependent on forest habitats for roosting and foraging. We analysed the habitat use of breeding colonies of Bechstein's bats in two wind parks. Over four years, we radio-tracked 31 individuals and identified their maternity roosts and foraging areas. We analysed the influence of the turbines on their habitat use, and the effect of wind speed and rotor blade rotation on the distance of the bats to the turbines. The colonies occupied tree roosts, a few hundred meters from the wind turbines. Foraging habitats close to the turbines were used preferably, when bats were close to the maternity roosts. The vegetation in these areas comprised of large trees and little shrub and herb layer coverage, indicating a high quality foraging habitat. The distance of the foraging bats to the turbines increased with increasing rotor blade rotation at high wind speeds. The results show that Bechstein´s bats become more selective in their habitat use, the closer they are to the wind turbines. Close to their maternity roosts, the advantages of a high quality habitat outweighed the disturbance effects and bats still used roosts and surrounding foraging habitats, despite turbine presence nearby. However, when further away from their roosts they avoided foraging close to wind turbines. With careful site planning, which excludes sensitive forest habitats, combined with restricted turbine operation times in summer, the negative effects of disturbances from wind turbines could be mitigated or avoided for maternity colonies of Bechstein's bats.

## Introduction

Worldwide, the renewable energy sector is growing immensely, to reduce carbon emissions and therefore mitigate climate change. Wind turbines, both onshore and offshore, are one of the most widely used forms of sustainable energy. While initially wind parks were installed in mostly open landscapes, in more recent years it has

**Data availability statement:** All relevant data are within the manuscript and its Supporting information files.

**Funding:** This study was funded by the Federal Agency for Nature Conservation (www.bfn.de) in Germany by means of the Federal Ministry for the Environment, Climate Action, Nature Conservation and Nuclear Safety (FKZ: 3515 86 1000). Members of both institutions took part in regular meetings of a project working group (PAG), where study design and results were discussed. The funders had no additional role in study design, data collection, decision to publish and analysis or preparation of the manuscript, but Frauke Krüger from the Federal Agency for Nature Conservation read the manuscript draft.

**Competing interests:** I have read the journal's policy and the authors of this manuscript have the following competing interests: Two of the authors (J.H, R.B.) work for the Freiburg Institute of Applied Animal Ecology, which is a consultancy agency, that also conducts wind farm impact assessment studies. Wind farm operators provided the turbine data for the data analysis. However the wind farm operators had no influence at all on study design, data collection, data analysis, interpretation and publication writing and editing. The authors take full responsibility for the integrity of the study. This does not alter their adherence to PLOS ONE policies on sharing data and materials.

become increasingly common to install wind parks in remote forest areas [1]. Wind turbines can have severe negative impacts on wildlife [2–4], the effects on bats in particular are widely documented. Wind parks in forests, a sensitive habitat for bats and other wildlife, must therefore be considered carefully. On the one hand, high flying bat species regularly die due to collisions with the rotating rotor blades or through barotrauma [5–7]. The increased mortality directly influences population sizes and can lead to negative population trends [8]. Furthermore, the installation of turbines in forests impairs or destroys the roosting and foraging habitats of several bat species [9–11]. At the wind turbine sites the forest habitats are lost completely due to logging, and bat colonies that roosted in these areas are forced to move. Moreover, there is growing evidence that also in neighbouring forest areas the habitat suitability for bats decreases, due to disturbance effects through noise or edge effects, which can lead to avoidance behaviour in bats. Avoidance effects can lead to further habitat losses and an increasing fragmentation of formerly undisturbed forest habitats. Several studies show that bat activity, especially that of narrow-space foragers, decreases in the close vicinity of wind turbines [12–19]. These studies are based on acoustic data, collected at the turbine sites and control sites [14,17], or along a distance gradient from the turbines [12,13]. Acoustic data of bat activity has several limitations, for example large differences in the detection distances for different species and a high dependence on site and weather conditions [20]. Additionally, acoustic data cannot provide information on the number of individuals present. Furthermore, most studies based on acoustic data lack information on bat colonies and roosts in the surrounding area. As such, it is difficult to predict the effect of a wind turbine on foraging behaviour, home ranges and the roost use of individual bats and local colonies solely from acoustic data. The effects of wind turbines on bats at the individual level have only been analysed in a single study to date, which used GPS-telemetry data from noctules (*Nyctalus noctula*), an open-space foraging bat [21]. The noctules did not avoid wind turbines in the direct surroundings of their roosts. However, avoidance behaviour occurred at other turbines at distances of 500 m to several kilometers away from their roosts, indicating that these bats avoid wind turbines during foraging. Narrow-space foraging bat species are especially sensitive to noise disturbances, as they use acoustic cues to find their prey. Therefore, they tend to avoid noisy environments [22,23]. Nevertheless, the direct effects of wind turbines on a narrow-space foraging bat species have not yet been studied.

The Bechstein´s bat (*Myotis bechsteinii*) is a typical forest bat, occurring mainly in Central Europe [24]. Their roosting areas are mostly located in old broad-leaved forests with large trees and a high proportion of dead wood and high densities of potential roost trees [25]. Bechstein´s bats almost exclusively use tree roosts as maternity roosts [26–29]. Similar to other forest bat species, they switch roosts often, sometimes daily [26,27,30] and split into subgroups, so a colony needs approximately 50 suitable and available roost trees within a small area [31]. Typically the females are philopatric and stay within the colonies they were born in [32]. Their foraging habitats are also located mostly inside forests. Bechstein´s bats always fly and forage close to vegetation, gleaning their prey from the foliage. They prefer closed forests

with different strata [26,30,33] or old forest stands without a shrub layer, where they hunt at ground level [24,34,35]. Their home ranges are small and when the juveniles are young, distances between the roosts and foraging habitats are normally less than 1 km [36,37]. Their core foraging areas are small, with average sizes ranging between 0.6 ha and 3.4 ha, and they are used territorially [27,38]. Due to their special habitat requirements Bechstein´s bats are an optimal model species for studying the effects of wind turbines in forests on a bat colony.

To analyse how the roost use and foraging habitat use of this narrow-space forager is affected by wind turbines in forests, we radio-tracked individuals from maternity colonies of Bechstein´s bats in two wind parks, situated in forests. We hypothesized (I) that the colonies would still use roosts and foraging habitats close to the turbines and (II) avoidance effects of the turbines would be mitigated by a high habitat quality where roost trees are close to the turbines. Furthermore, we expected (III) that avoidance behaviour would increase with increasing rotor blade rotation. Based on the results of this study, we give recommendations for future wind park planning with respect to planning considerations and mitigation for forest bats.

## Materials and methods

### Study area and surveys before complete wind turbine installation

The study took place in two wind parks in the Rhineland-Palatinate, one in Hunsrück (HR) and the other in Palatinate (PL) (Fig 1), both with known colonies of Bechstein´s bats. The study area HR lies in the western part of the low mountain range Hunsrück, close to the river Moselle at an altitude of approximately 400 m. The area is covered with forests of different types and ages and agricultural fields. The wind park consists of 15 turbines with a rotor diameter of 70 m and a total height of 149 m, which were installed between 2004 and 2006, and one further turbine with a rotor diameter of 82 m built in 2011. Eight turbines were placed inside the closed forest, in areas that are dominated by younger coniferous trees. The colonies of Bechstein´s bats were surveyed in 2007. The bats used habitats adjacent to the turbines, in areas with old broad-leaved trees [39]. In total there were a minimum of 82 individuals (including juveniles), which roosted in different trees; whether they belonged to subgroups of one colony or several colonies could not be

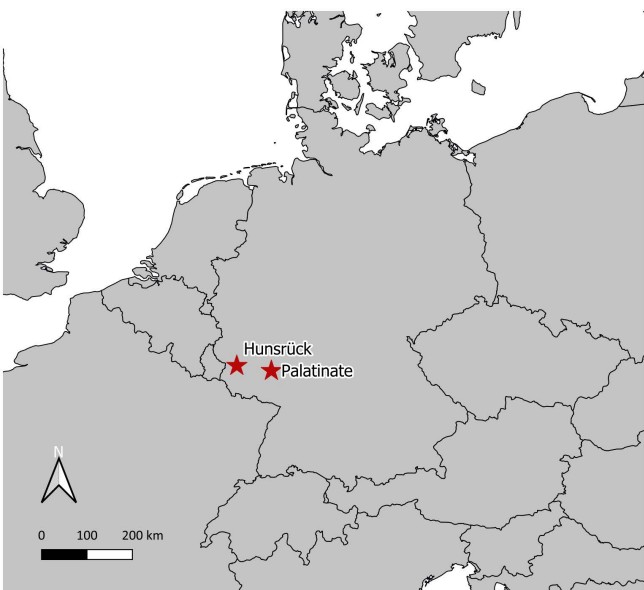

**Fig 1. Study areas in Europe (Background map: naturalearthdata.com).**

clarified. Turbine operation was restricted to prevent collisions with bats. The study area PL lies in the low mountain range area Saar-Nahe-Bergland, at an altitude of 300 m. Large parts of the area are covered with deciduous trees of different ages. In the area, old hall-like forests alternate with areas of succession and bush land as well as agricultural land. Old military buildings are located in the middle of the study area. This wind park consists of three turbines, with a rotor diameter of 126 m and a total height of 212 m, which were installed in 2019. The turbines were placed in areas of young successional forest growth. Additionally, there are turbines in the adjacent fields. Bechstein´s bats were surveyed in 2012 in PL, were they roosted in trees in an old forest area next to one of the planned turbine locations [40]. In total there were a minimum of 17 individuals, which all roosted in one tree. The turbines were approved with mitigation measures to compensate for habitat losses and a restricted operation regime to prevent collisions with bats. In both study areas, the wind turbines are operated with turbine specific curtailment algorithms [41], which are based on the results of acoustic monitoring at the nacelle of the turbines and determine the cut-in wind speed, depending on the time of year and the time of night. In PL the curtailments are comparatively strict, as the turbines are big and maternity roosts of several species of collision risk were found in the pre-study [40].

### Field methods

**Radio tracking.** In both study areas the field work took place in two seasons, in 2019 and 2022 in HR and in 2020 and 2023 in PL. The bats were radio-tracked during maternity season, shortly after the birth of the juveniles, from the end of June until the end of July. To catch female Bechstein´s bats, we installed approximately 10 mist nets (6−12 m, ecotone, Gdynia, Poland) in the surroundings of known habitat trees of the maternity colonies. Only adult female bats, which were lactating or in few cases not reproductive, were chosen for radio tracking. We used small VHF-transmitters (V3, Plecotus Solutions, Freiburg, Germany) with a weight of 0.35 g and an operating time of approximately 10 days. The transmitters weight was below 5% of the bats' body mass, which is a common threshold for radio tag weights in bats [42]. The transmitters were glued into the fur on the back of the tagged individuals, using a medicinal skin glue (Manfred Sauer GmbH, Lohbach, Germany). The transmitter signal was tracked by TRX 1000S receivers (Wildlife Materials Int., Murphisboro, USA) and three-element yagi antennas (Biotrack LTD, Wareham, UK). In total, 31 females were radio tracked, including 29 lactating and two non-reproductive females (Table S1 in S1 File). Each individual was radio tracked for two to four nights. This short tracking time is appropriate for Bechstein´s bats which show strong territoriality and site fidelity [27,38]. Individuals were tracked from roughly two hours after sunset until half an hour before sunrise. Cross bearings were taken by two observers simultaneously in three minute intervals. The tracking was interrupted when the transmitter signal did not change position within the three minute interval and it could be assumed that the bat was stationary. Resultantly, each stationary position was only tracked once. The observer positions were selected to be as close as possible to the used foraging habitat, to minimise bearing errors. The positions were recorded via GPS with the program QField on a tablet (SM-T280 and SM A14). To synchronize the timings of the bearings the observers were in contact via walkie-talkies, the direction of the strongest signal was recorded with a compass. The day roosts of all radio tracked animals were searched for daily. We carried out emergence counts, sometimes at several roosts at the same time, to determine the number of individuals in the area. Due to the strong fission-fusion behaviour of these bats, as noted in the pre-studies, it was not possible to determine whether the tracked bats belonged to subgroups of one colony or several colonies. Permits for working with bats were granted by the local authorities (Struktur- und Genehmigungsdirektion Süd and Nord, Rhineland-Palatinate, permit numbers # 42/553–254/56–20, # 425-107-235-0002/2019, #425-107-235-0001/2022).

**Mapping of habitat structure.** We mapped the vegetation structure inside the home range of the colonies, calculated as a minimal convex polygon (MCP) of all tracking points with a buffer of 50 m. The mapping in the autumn of the first study year took place in the complete home range, based on the radio tracking results of the first study year. In the autumn of the second year, mapping only took place in parts of the home range which had not yet been mapped. For

the mapping, we initially defined vegetation units with similar forest structure using aerial pictures (Google Earth and GeoBasis DE-BKG). We conducted one point and one 50 m transect count of vegetation per hectare. Points and transects were equally distributed within the vegetation units, with a minimum distance of 10 m to the unit borders. In the study area HR, the home range included 396 mapping points and transects, in PL home ranges included 260 mapping points and transects. Along the 50 m mapping transects all trees with a diameter at breast height (DBH) of more than 10 cm and within a distance of 5 m from the transect were counted. For each of these trees, we identified the species and recorded the DBH. To measure the distance of the trees to the transect line, we used a laser distance meter (Leica Disto D5/D510). In a radius of 10 m around each mapping point, we determined the coverage of leaf layer, herb layer, bush layer and tree layer vegetation (in the categories 0–10%, 10–20%, 20–40%, 40–60%, 60–80%, 80–100%).

## Turbine data

Data on wind speed (m/s) and temperature (C°) at nacelle height and rotor blade rotation (rpm) were provided by the turbine operators for our analyses. This data was recorded in 10 minute intervals. In the study area HR, data from 15 of 16 turbines were available, in PL the data of all three turbines were available.

## Data analysis

For all analyses we used the programs R 4.2.2 (R Foundation for Statistical Computing) and QGIS 3.34.4 (Open Source Geospatial Foundation Project). We computed the positions of the bats by calculating the intersections of each pair of bearings. Positions between the two bearings with angles of less than 30° and more than 150° were excluded due to large uncertainties in the calculated positions. We excluded the stationary positions of bats when it was likely that they were not linked to foraging, but nursing behaviour (which was the case when bats were inactive for approximately more than 10 minutes in the area of the known maternity roost tree). Furthermore, we excluded positions outside of the forest, as Bechstein´s bats do not forage in open landscapes and those points represented transfer flights or were caused by bearing errors. The final data set consisted of 3,413 fixes.

   The roosting area and the home range of the bats were calculated as the MCP of all radio fixes (i.e., positions) [38,43]. We calculated the 95%- and 50%-Local Convex Hulls (LoCoHs) [44] as individual foraging search areas (95%-LoCoHs) and core foraging areas (50%-LoCoHs) [38]. This method does not smooth location probabilities and therefore „strong" boundaries, e.g., between forest and open land are more aptly accounted for. For the calculation we used the R-package AdeHabitat with the a-LoCoH-algorithm [45], where a was determined for each individual as the largest distance between two positions [38]. With this algorithm the kernels are constructed from all points within a radius a such that the distances of all points within the radius to the reference point sum to a value less than or equal to a. For two individuals it was not possible to calculate the 50%-LoCoHs due to a lack of sufficient data points and positions which were too scattered.

   We used a conditional logistic regression model to analyse how habitat preferences of Bechstein´s bats are influenced by the distance to the turbines. For the response variable we used pairs of presence (value 1) and pseudo-absence (value 0) points. Presence points were radio fixes, i.e., positions in habitats which have been used by bats. Pseudo-absence points were randomly drawn from the set of regular habitat mapping points within the individual foraging search areas (95%-LoCoHs). As the Bechstein´s bat is territorial regarding its foraging areas [38], we assumed this to be the most realistic approach to determine the available habitat for the individuals. For each presence point, we allocated the vegetation parameters of the closest mapping point within the same vegetation unit. We created 10 data sets with random pairs of all fixes of the final data set (presence points) and pseudo-absence points, to estimate how strongly the result is influenced by the random pairing of presence and pseudo-absence points. Pairs, that had at least one habitat variable missing in one of the two points, were removed from these data sets. Thus 3,392–3,398 pairs remained in the 10 data sets. Furthermore to reduce autocorrelation and to keep computing time reasonable, we reduced the data set and used only every third radio

fix. As we were interested in habitat preferences averaged across all individuals, the loss in precision of the estimates is negligible, when reducing sample sizes within individuals.

Prediction variables in the regression were habitat variables, which were expected to have an ecological meaning for Bechstein´s bats due to their biology. Where variables were strongly correlated, only the variable with the more logical ecological meaning was used. Correlations occurred between leaf cover and herb cover (r = 0.81), leaf cover and the proportion of broad-leaved trees (r = 0.49) and the proportion of oak and the proportion of broad-leaved trees (r = 0.45) – we selected the variables herb cover, as ground coverage is important for Bechstein´s bats, which use to forage close to the ground [34,46], and the proportion of broad-leaved trees, as Bechstein´s bats occur in different types if broad-leaved trees [47]. For the final model we used the following habitat parameters: Tree cover, shrub cover, herb cover, proportion of broad-leaved trees, median of DBH, standard deviation of DBH, distance to the nearest turbine and distance to the currently used day roost. Where bats changed their day roosts, we used their evening roost for all fixes before midnight and the roost, which was used the following morning for all fixes after midnight, as we assumed that the bats changed roosts during the night. To analyse the impact of the wind turbines on habitat choice, we included the interaction between the distance to the next turbine and all other habitat variables. Habitat preferences were calculated for each individual, accordingly the individual ID was set as a random effect. We included normally distributed, individual-specific effects for each habitat variable. We did not include an individual-specific intercept, because the conditional logistic regression does not estimate the intercept. Rather the intercept is defined by the design, that means by the proportion of used points in each stratum. In our case, a stratum is a pair of one used (presence) and one available (pseudo absence) point.

To fit the model we used the function stan_clogit in the R package rstanarm, which is based on Bayesian statistical methods. The function calls the software Stan [48] which uses Hamiltonian Monte Carlo [49,50] to fit the model to the data. Four chains with 2,000 iterations were simulated and from each chain the last 1,000 values were used. In total 4,000 draws were used to describe the posterior distributions of the model parameters. The convergence of the simulation was assessed based on the R̂ value and the number of effective independent draws [51]. For the final model, the 4,000 simulated values from each of the 10 data sets were joined and used to describe the averaged effects over all data sets. The mean values were used as an estimate and the 2.5% und 97.5% quantiles of the A-posteriori distribution were used as the upper and lower limits of the 95% credibility intervals. The residuals did not show any autocorrelation.

In a second model we analysed whether wind speed and rotor blade rotation influences the distance of the bats to the turbines. We matched the wind speed and rotor blade rotation from the nearest wind turbine to each radio fix using the timestamp. Where such data were not available for the closest wind turbine, this fix was excluded from this analysis. In total 145 radio fixes had to be excluded for this reason, reducing the dataset to 3,253 radio fixes. In a linear mixed model, we used the distance to the turbines as the response variable and wind speed, rotor blade rotation, distance to roost and study area as prediction variables. The two and three-fold interactions of variables wind speed, rotor blade rotation and study area were also used as further prediction variables. To account for repeated measurements of the same individuals, we included the individual ID as a random factor. Model fit was assessed by visually inspecting the distribution of the residuals (see S2 File).

## Results

### Roost use

In the study area HR, we found 14 roost trees in total, six in 2019 and nine in 2022. One tree was used in both years (Fig 2 and Table S2 in S1 File). The size of the roosting area was 19.6 ha in 2019 and 22.3 ha in 2022. The distance between two roosts ranged between 11 m and 1,194 m (mean 507 m ± 275 standard deviation, hereafter sd) and distances between roosts and turbines ranged from 150 m to 811 m (mean 415 m ± 225 sd). In comparison to the results of the survey in 2007, where the distance from the roosts to the turbines ranged between 90 m and 1,274 m (mean 620 m ± 560 sd), the core roosting area had moved to the north, but not further away from the wind turbines. While in 2019 most of the roosts

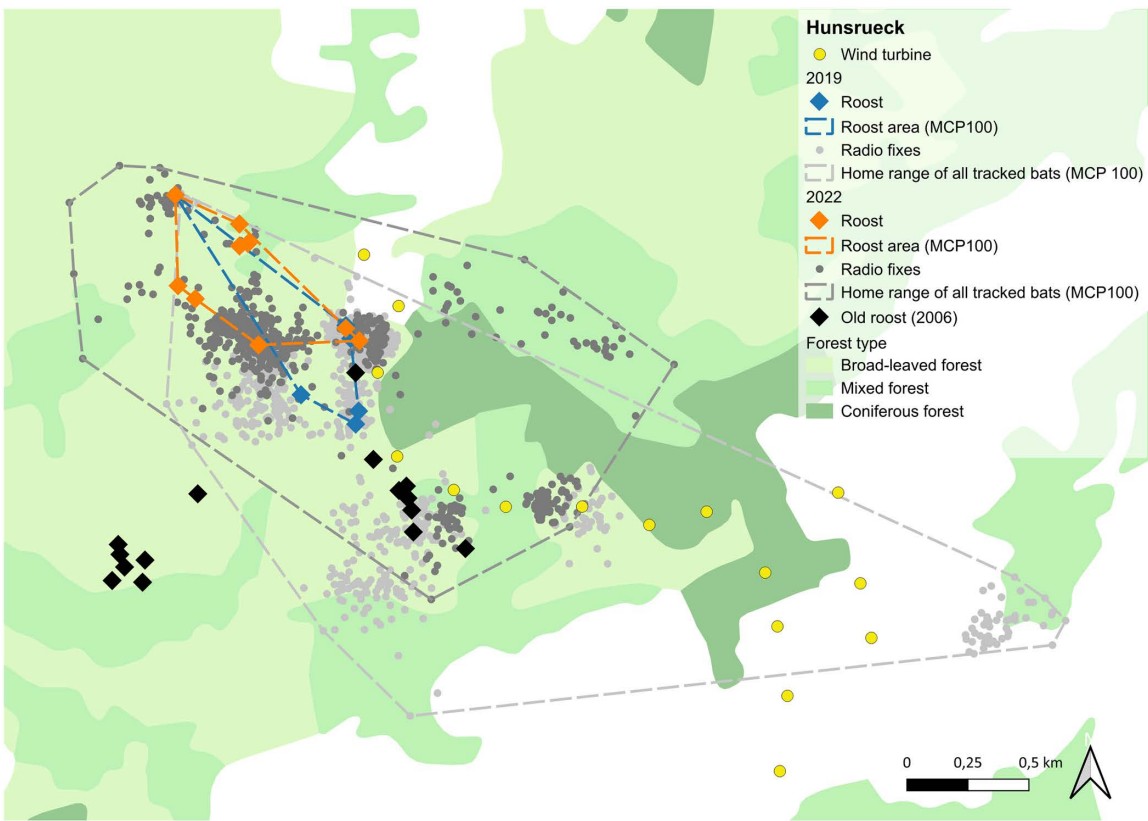

**Fig 2. Results of the radio tracking in Hunsrück: Roosts are shown as squared dots in orange for 2019 and blue for 2022 and radio fixes of all animals are shown as small light grey dots for 2019 and dark grey dots for 2022.** MCP 100 areas of roosts and radio fixes are shown as dashed lines in the same colors. Yellow dots indicate wind turbines. Background Map: Corine Land Cover 5 ha, GeoBasis-DE/BKG 2018, dl-de/by-2-0 (www. govdata.de/dl-de/by-2-0).

were located in the direct surroundings of the turbines, in 2022 more roosts were used further away from the turbines in the north west of the study area.

In the study area PL, we found 26 roost trees in total, 11 in 2020 and 15 in 2023 (Fig 3 and Table S2 in S1 File). The size of the roosting area was 57.2 ha in 2020 and 105.2 ha in 2023. The distance between two roosts ranged between 22 m and 2,020 m (mean 852 m ± 444 sd) and distances from the roosts to the turbines ranged from 70 m to 684 m (mean 295 m ± 161 sd). The roosts were distributed through the entire wind park, in 2020 they were mostly in the north and south and in 2023 more in the east and west. An area with a hall-like old forest next to a turbine, where roosts were found in 2012, was still intensively used by the bats in both study years.

The number of days, during which the bats used the same roost, varied strongly in both areas. In HR the roosts were used for one to ten days (mean value 3.06 days ± 3.09 sd) and in PL for one to 13 days (mean value 3 days ± 3.43 sd) (Table S3 in S1 File). The number of bats in a roost also varied. In HR in 2019 the maximum number of bats in a roost was 51 and the total number of bats at parallel emergence counts was 80. In 2022 the maximum number of bats in one roost was 61 and at parallel counts 88 bats were recorded (Table S3 in S1 File). In PL in 2020 the maximum number of emerging bats from one roost was 58 and 59 were counted at parallel counts. In 2023 the maximum number at one roost was 30 and 44 individuals were recorded at parallel counts.

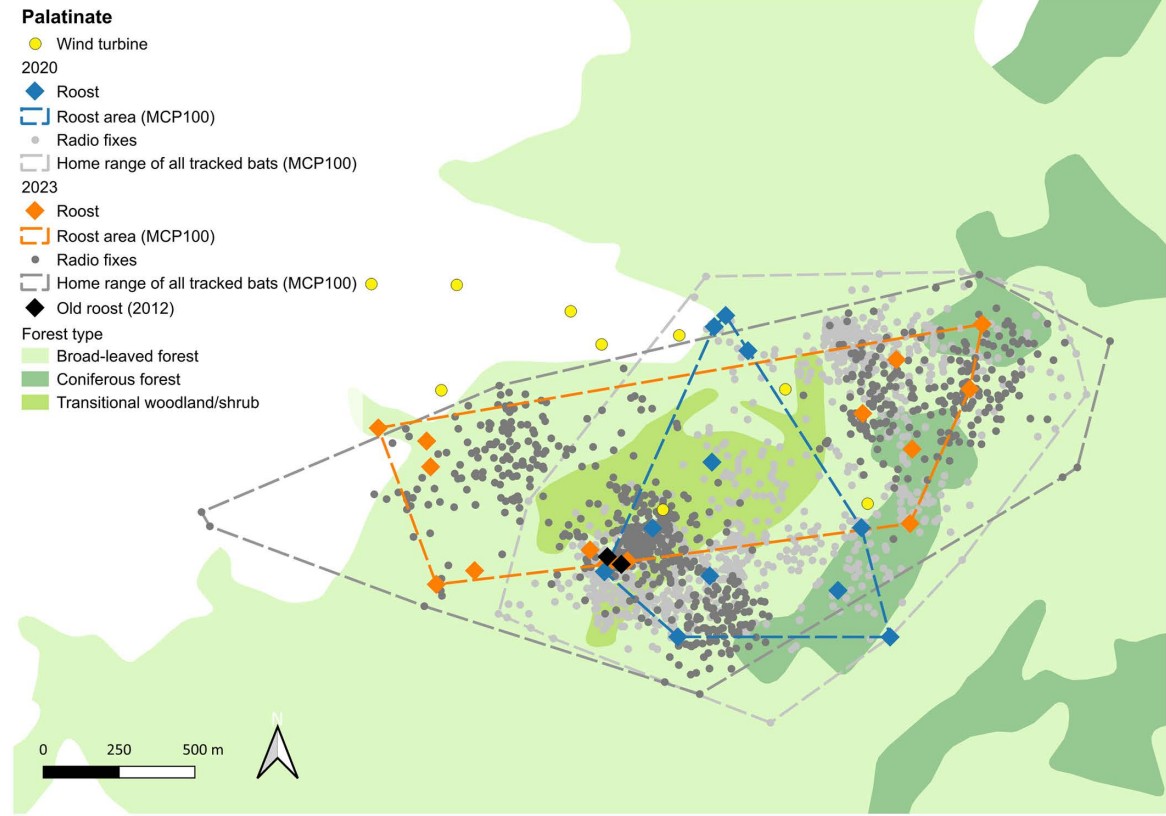

**Fig 3. Results of the radio tracking in Palatinate: Roosts are shown as squared dots in orange for 2020 and blue for 2023 and radio fixes of all animals are represented by small light grey dots for 2020 and dark grey dots for 2023.** MCP 100 areas of roosts and radio fixes are shown as dashed lines in the same colors. Yellow dots indicate wind turbines. Background Map: Corine Land Cover 5 ha, GeoBasis-DE/BKG 2018, dl-de/by-2-0 (www.govdata.de/dl-de/by-2-0).

## Foraging habitat use

The size of the home ranges (MCPs) of all tracked bats in HR was 384.58 ha in 2019 and 259.34 ha in 2022 (Fig 2) and in PL 187.65 ha in 2020 and 222.89 ha in 2023 (Fig 3). The size of the home ranges of the individuals was between 6.59 und 172.05 ha (mean 40.36 ha±44.56 sd) in HR and between 10.22 and 92.27 ha (mean 38.78 ha±23.92 sd) in PL (Table S4 in S1 File). Most individuals stayed close to their roosts: in HR 49.8% of all fixes were located within 500 m from the roosts and 89.4% were within 1,000 m of the roosts. In PL 65,6% of the fixes were located within 500 m from the roost and 94.1% were within 1,000 m (Fig 4). The maximum distance between the fix of an individual and the roost it was using was 3.74 km in HR and 1.8 km in PL respectively.

The size of the 95%-LoCoHs, which are interpreted as foraging search areas, varied strongly and ranged between 2.14 and 32.16 ha in HR (mean 9.58 ha±8.77 sd) (Fig 5 and Table S4 in S1 File) and 4.30 and 32.20 ha in PL (mean 14.15 ha±7.18 sd) (Fig 6 and Table S4 in S1 File). The individuals used between one and three search areas during the tracking period. Typically one of these was located close to the roost, while the others were located further away. The size of the core foraging areas (50%-LoCoHs) ranged between 0.55 und 5.90 ha (mean 1.85 ha±1.82 sd) and in HR and between 0.87 und 6.95 ha (mean 2.78 ha±1.70 sd) in PL. Most individuals had only one core foraging area. The core areas overlapped partially, mostly close to the roosts.

The habitat preference analysis showed that the Bechstein´s bats preferred to remain close to their roosts when turbines were present in the close vicinity (Table 1 and Fig 7C). They used habitats close to the turbines particularly when

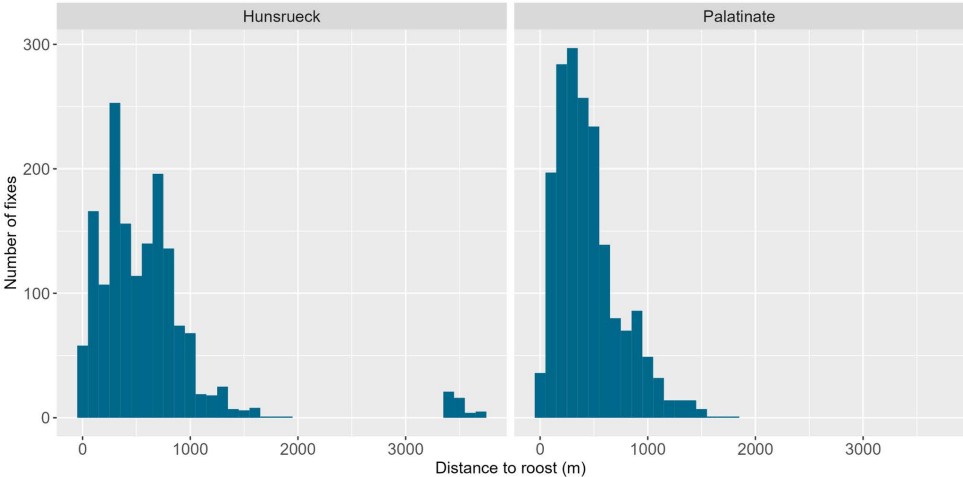

**Fig 4. Distribution of the distances of all radio fixes to the roosts.** The graph on the left shows the results for Hunsrück (n = 16,00), the graph on the right for Palatinate (n = 1,813).

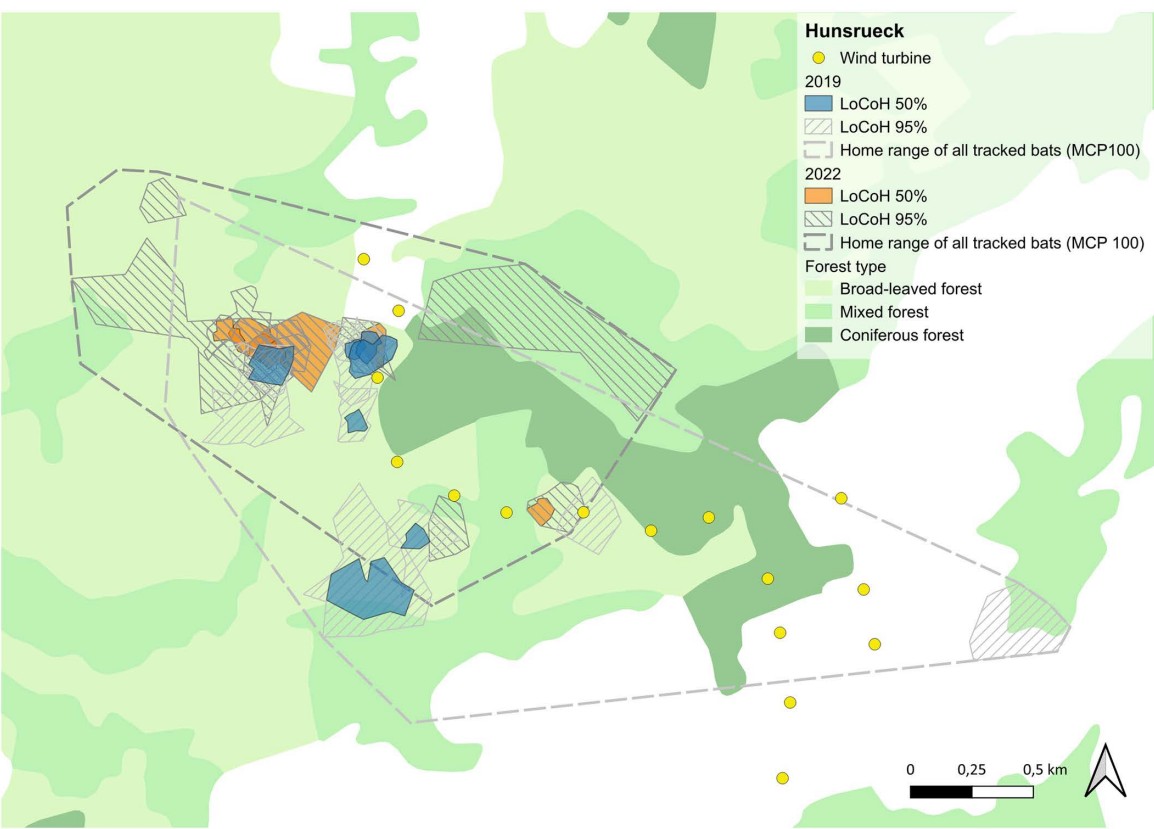

**Fig 5. Foraging areas in Hunsrück: 95%-LoCoHs (Foraging search areas) are displayed as hatched light grey areas for 2019 and hatched dark grey areas for 2022, 50%-LoCoHs (Core foraging areas) are shown in blue for 2019 and in orange for 2022.** Yellow dots indicate wind turbines. Background Map: Corine Land Cover 5 ha, GeoBasis-DE/BKG 2018, dl-de/by-2-0 (www.govdata.de/dl-de/by-2-0).

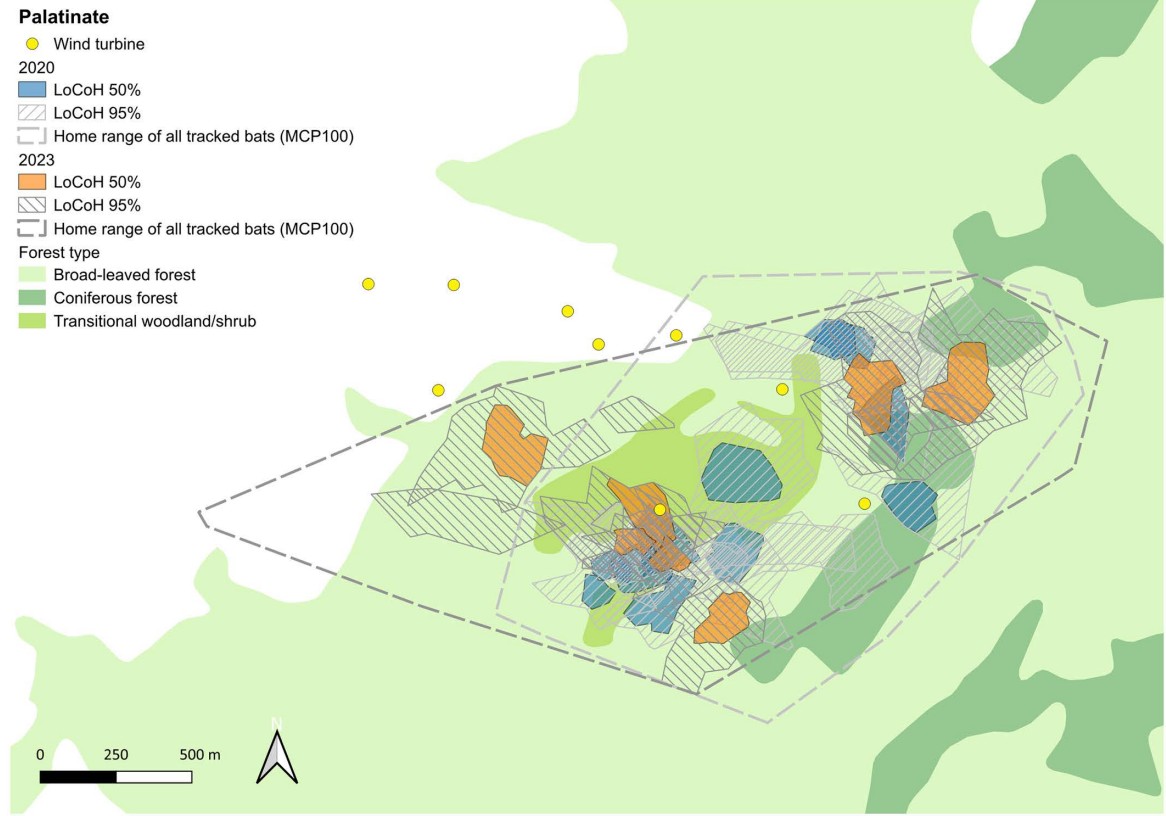

**Fig 6. Foraging areas in Palatinate: 95%-LoCoHs (Foraging search areas) are displayed as hatched light grey areas for 2020 and hatched dark grey areas for 2023, 50%-LoCoHs (Core foraging areas) are shown in blue for 2020 and in orange for 2023.** Yellow dots indicate wind turbines. Background Map: Corine Land Cover 5 ha, GeoBasis-DE/BKG 2018, dl-de/by-2-0 (www.govdata.de/dl-de/by-2-0).

they were in the direct surroundings of their roosts. Furthermore, in small distances from the turbines the bats showed a preference for habitats with a higher median DBH (Fig 7E). A high coverage of shrub and herb layer (Fig 7B and 7G) was avoided close to the turbines. There was no clear effect of the proportion of broad-leaved trees, the tree layer coverage and the standard deviation of the median of the DBH (Fig 7A, 7D and 7F.

In both study areas, the bats were active up to high wind speeds at nacelle height of 8 m/s in HR and 12 m/s in PL (Table 2 and Fig 8). A correlation between the distance of the bats to the turbines and rotor blade rotation was only present at high wind speeds. In HR the distance of the bats to the turbines increased with increasing rotor blade rotation at wind speeds of 8 m/s. In PL this correlation was only observed at wind speeds of 12 m/s. The data shows a clear difference between the wind parks in their operating schemes. While in HR the turbines started to run with a relatively low rotor blade rotation of only 5 rpm and therefore at low wind speeds, in PL the turbines began to operate at high rotor blade rotations, close to full speed representing high wind speeds. In both cases, rotor blades were still moving at a very low speed when the turbines were not operating.

## Discussion

### Roost use

In both study areas, Bechstein´s bats did not avoid roosts close to the turbines. In HR the roosting area had moved several hundred meters further north, compared to the survey 13 years earlier. However, despite the turbines being in

**Table 1. The results of the habitat preference model, dependent on the distance to the wind turbine: Estimated values of the a-posteriori-distribution of the model parameters and 95% credibility intervals. Values shown are the mean values for the 10 data sets. Prediction variables with colons indicate interactions of two prediction variables. WT stands for wind turbine. A negative value of the main effects mean that bats preferred foraging at lower values of the respective habitat variable, whereas positive values mean that they preferred higher values. The interaction parameters show how the preference (log-odds) changed with distance to the wind turbine (WT). The regression lines defined by the main effects and the interactions are shown in Fig 7.**

| Prediction variable | Estimated value | 95% lower limit | 95% upper limit |
|---|---|---|---|
| Distance to WT | −0.97 | −1.78 | −0.25 |
| Distance to roost | −3.25 | −4.80 | −2.13 |
| Herb cover | −0.14 | −0.45 | 0.14 |
| Shrub cover | −0.04 | −0.52 | 0.38 |
| Tree cover | 0.07 | −0.37 | 0.52 |
| Proportion of broad-leaved trees | 0.17 | −0.33 | 0.71 |
| Median of DBH | −0.09 | −0.52 | 0.33 |
| SD of DBH | 0.60 | 0.10 | 1.25 |
| Distance to WT: Distance to roost | 0.94 | 0.50 | 1.42 |
| Distance to WT: Herb cover | 0.18 | −0.19 | 0.52 |
| Distance to WT: Shrub cover | 0.09 | −0.19 | 0.37 |
| Distance to WT: Tree cover | −0.12 | −0.59 | 0.27 |
| Distance to WT: Proportion of broad-leaved trees | −0.10 | −0.45 | 0.24 |
| Distance to WT: Median of DBH | −0.16 | −0.48 | 0.17 |
| Distance to WT: SD of DBH | −0.04 | −0.37 | 0.30 |

operation for over 15 years, the new roosting area was still located close to the turbines, with turbines at a minimum distance of 150 m from roosts. The most probable reason for this shift in roosts are changes in the available roosts, despite no visible evidence of an altered forest structure in the old roost areas. In PL, the roosting area close to the turbines, which was identified in the pre-construction survey eight years earlier, was still used intensively by the bats.

The size of the roosting areas was different in both study areas and years, varying in size between 19 and 105 ha. Variation in roosting areas are not unusual for Bechstein´s bats and were also found in other colonies in Germany and Luxembourg [37]. It is possible that the tagged bats belonged to different colonies or sub-colonies, since tagged bats didn´t always share roosts and some individuals exclusively used groups of roost trees, which were never used by the other tagged individuals. It is well known that roosting areas and home ranges of two colonies of Bechstein´s bats can overlap noticeably [37] and that they can build subcolonies, between which only single individuals change from time to time [52]. Within the short study period and without further genetic analyses, the colony structures cannot be analysed in detail.

The animals showed a typical fission-fusion behaviour with frequent roost changes and changing group sizes [31]. The bats changed roost on average every three days. Yet, particularly at the beginning of the study period, shortly after the birth of the juveniles, roosts were used for up to 10 days in a row. Presumably, it is a big effort for the mothers to move their young to a new roost as long as they have not yet fledged [30,53]. The group sizes changed between one und 58 animals. The maximum numbers of individuals counted on the same evening simultaneously at different roots was 59 individuals in PL and 88 in HR, including fledged juveniles. Colony sizes of Bechstein´s bats are documented to be between 10 and 50 individuals, but they can rarely reach numbers of up to 100 [47]. Accordingly, the number of individuals in the study areas were remarkably high, indicating that the tracked bats could belong to more than one colony. In the study area HR in the survey in 2007, shortly after the turbines were built, the maximum number of animals was 82. Although the roosting area changed, the individual bat numbers in the two survey years were similar to the survey from 2007 (15 years earlier). During the preconstruction survey in PL in 2012 there were only 17 animals counted, of which only one was tracked. We presume, that these 17 bats were

**Fig 7. Results of the habitat preference model: Influence of the distance to the turbines on the preference of different habitat parameters, based on data from all sites and years.** The blue translucent areas show the 95% credibility intervals for the ten different data sets. The black line depicts the mean estimated value over the data sets. A preference value of 1, shown with a grey line, indicate that a structure is used according to its availability, higher values indicate a higher than average use and lower values a lower than average use.

a subgroup of the colony. Therefore, a comparison of the number of individuals before and after construction is not possible. As individual numbers after construction were quite high for colonies of Bechstein´s bat, there was no indication that populations have decreased since turbine construction. Nevertheless, negative effects on the local population, for example a reduced juvenile fitness or an increased mortality cannot be ruled out. It can take a long time for a negative population trend to become obvious and due to the fission-fusion behaviour, it is difficult to get reliable count data from forest dwelling bats. Reduced population numbers could be compensated by immigration from neighbouring colonies; however this is unlikely for Bechstein's bats, as they live in closed societies and immigration events are very rare [54]. Furthermore, there could be a threshold of turbine density that colonies can tolerate inside their habitats. Long-term-studies on population trends of Bechstein colonies close to wind parks, of varying sizes and at reference sites, would be necessary to answer these questions.

**Table 2. Model results of the effects of wind speed and rotor blade revolution on the distance to the turbines: Estimated values of the a-posteriori-distribution and 95% credibility intervals. Prediction variables with colons indicate interactions of two prediction variables.**

| Prediction variable | Estimated value | 95% lower limit | 95% upper limit |
|---|---|---|---|
| Intercept | 428.59 | 362.97 | 493.58 |
| Wind speed | −1.93 | −18.18 | 14.59 |
| Rotor blade revolution | −38.34 | −52.07 | −24.14 |
| Distance to roost | 29.83 | 24.34 | 35.35 |
| Study area Palatinate | −236.61 | −262.61 | −210.71 |
| Wind speed:Rotor blade revolution | 64.22 | 52.94 | 75.47 |
| Wind speed:Study area Palatinate | 12.63 | −6.72 | 32.12 |
| Rotor revolution:Study area Palatinate | 5.32 | −17.05 | 28.06 |
| Wind speed:Rotor blade revolution:Study area Palatinate | −39.03 | −53.31 | −24.62 |

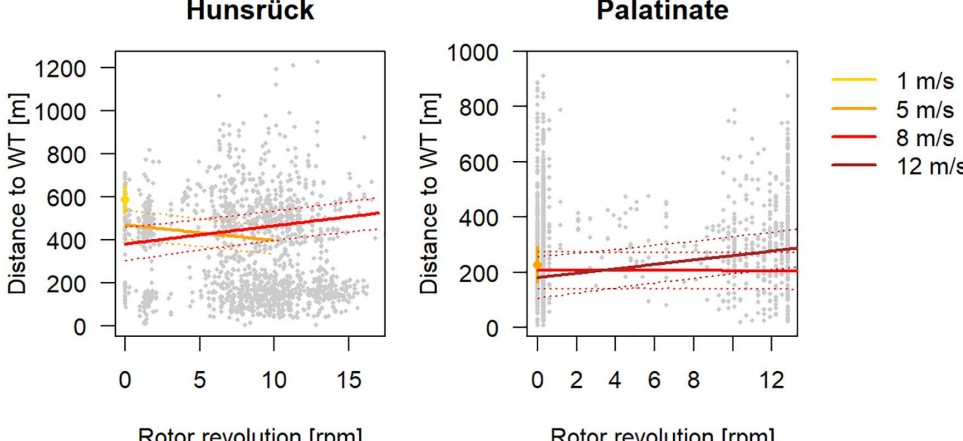

**Fig 8. The effect of rotor blade revolution on the distance of bats to the next turbine for different wind speeds.** The mean distance of bats to the turbine for wind speeds of 1 m/s (Hunsrück) and 5 m/s (Palatinate) are only shown for a rotor blade revolution of 0 rpm, as the turbines started operating at higher wind speeds. The grey dots show the distribution of the raw data.

## Foraging habitat use

The radio-tagged bats showed typical patterns of habitat use, known from other studies on Bechstein´s bats. The size of the home ranges of the tagged bats was between 188 and 386 ha. This is within the normal range for colonies of Bechstein´s bats, for which sizes of up to 500 ha are documented [38]. The size of the individual foraging search areas varied strongly and ranged between 7 and 172 ha, although they tended to be smaller when close to the roosts. The core foraging areas were relatively small and ranged in size between 0.7 and 7 ha. This resembles the typical foraging behaviour of Bechstein´s bats, which often forage for a long time in a small area [34]. Individual foraging habitats were mostly located within 1 km of the roosts. Especially after the females have given birth to their young, Bechstein´s bats only cover short distances between roosts and foraging habitats, as this enables efficient foraging flights with low energy consumption [27,30,37]. Nevertheless, core foraging habitats only had small overlaps reflecting the territorial hunting behaviour of the Bechstein´s bat [35,55]. One individual was recorded at a distance of 4 km to its day roost, which is quite far for a Bechstein´s bat, but has also occurred in other studies especially when the juveniles were older [35,37].

Habitat preferences of the Bechstein´s bats changed in relation to the distance to the turbines. Foraging habitats close to the turbines were only used close to the roosts and were avoided with increasing distances to the roosts. Furthermore, in areas closer to the turbines, bats showed a preference for habitats characterised by larger tree DBH and reduced herb and shrub layer coverage. These parameters resemble a typical high quality habitat for Bechstein´s bats with old trees, a closed tree layer and little ground vegetation, where they can forage close to the ground on the foliage cover [34,46]. Thus for the Bechstein´s bat, the advantages of a high quality habitat close to the roosts seemed to outweigh the disadvantages of disturbances due to turbine operation. Furthermore a study by Sotillo, le Viol [56] was able to show, that a high habitat quality next to turbines can mitigate the negative disturbance effects. On the other hand, the Bechstein´s bats clearly avoided foraging in habitats further away from their roosts, when these habitats were close to wind turbines. As a consequence, the habitat availability in the more distant surroundings of the roosts is decreased by turbine operation, as disturbance from the turbines leads to these otherwise seemingly suitable habitats not being used. This avoidance behaviour was also present in acoustic studies, in which the activity of bats, categorized as narrow-space hunters, decreased with decreasing distance to the turbines [12,14–16]. Our study focussed on the time of year when the juveniles are still young and immobile and adults are forced to stay close to their roosts. It is possible that the avoidance behaviour of Bechstein´s bats seen in this study also extends to habitats close to roosts outside of the time period we studied (the lactating phase), when individuals can regularly cover longer distances. Studies with several tracking-periods in different seasons of the year would be appropriate for gaining a better understanding of avoidance behaviour throughout the year. Furthermore, Bechstein´s bats show a strong site fidelity [31,55], which could make them less flexible to react to changes inside their habitats. It is possible, that other forest bat species react differently, for example with site shifts or a stronger avoidance behaviour, when wind turbines are installed in their home ranges. On the other hand, other bat species, which forage along vegetation lines or in open space, for example bats of the genus *Pipistrellus*, can be attracted to wind park sites, resulting in a higher collision risk [57]. Further research is needed for a better understanding of the extent and importance of the avoidance behaviour of forest bat species.

The distance of the animals to the turbines increased with increasing rotor blade rotation, which is linked with increasing noise emissions. This indicates a reaction of the bats to the disturbance effects of the operating turbines, as it is also supported by Ellerbrok, Farwig [13] based on acoustic data. Disturbance effects of noise are already documented for bats of different species [22,23,58,59]. In HR this relationship was already present at wind speeds of 8 m/s and in PL of 12 m/s. We assume, that the differences in turbine types and curtailment regimes in both wind parks led to variable noise patterns. In PL the curtailment regime is quite strict due to the turbine size and the presence of maternity roosts of species with high collisions risk [40]. As turbines in HR have to be curtailed at medium wind speeds, it´s possible that a disturbance effect is only present there at high wind speeds. Hence, strict curtailments may not only prevent collisions, but could also have positive effects on the use of habitats close to wind turbines.

## Implications for wind park developments

Our results show, that maternity colonies of Bechstein's bats are able to use roosts and foraging habitats intensively, though these were located in the direct surroundings of wind turbines. We did not find any indication of considerable negative effects of the studied wind turbines, like severe changes in the home range of bats or a decreasing number of individuals, despite the fact that one of the studied wind parks has been in operation for over 15 years. We recommend that pre-construction studies focus on identifying core colony roosting areas, as the protection of these areas is essential for the continuity of the colonies. To determine these core areas, it is necessary to track several individuals of a colony during the maternity season, as forest-dwelling species switch roosts on a regular basis. As a mitigation measure, these core areas should stay free from turbines, ideally with an additional buffer of several hundred metres. This mitigation measure should be the minimum for species with small home ranges like Bechstein´s bats, in order to protect the most important

foraging habitats in the direct surroundings of the roosts. Additionally, the habitat loss inside the impact areas, caused by the logging of trees must be mitigated, for example by identifying forest protection areas in the close surroundings.

Although this study shows, that maternity colonies of Bechstein´s bats are able to cope with the construction and operation of wind turbines within their home range, it is important to keep the avoidance behaviour in mind, which was shown in this study for forest areas further away from the roosts. Similar behaviour was detected in several other studies based on acoustic data [12,14,15]. In general, further research is needed, to better understand this behaviour. Principally it can be assumed, that the operation of wind turbines in forests leads to disturbance effects, which reduce the habitat suitability of forest areas close to the turbines. This may lead to a lower foraging activity of bats, furthermore it may prevent bats settling in new habitats with otherwise good quality when they are in the vicinity of wind turbines. Therefore, it is still important to concentrate wind energy in less sensitive areas: ideally wind parks should be located outside of forests, as also recommended by EUROBATS [60]; where this is not feasible, wind parks should be planned in clear cuts and succession areas, with low habitat quality for bats. Important forest areas like broad-leaved forests, mature forests over 80 years old, natural boreal forests and forests in Natura 2000 areas should stay free from wind turbine planning.

## Conclusion

For the present study the habitat use of Bechstein´s bats at two study sites was analysed. The extent to which these results can be transferred to other colonies, other species and other areas is limited. More studies in other areas, over longer time periods and with other bat species are important to strengthen and verify the results.

The study results give an opposing picture: On the one hand, the studied Bechstein´s bats can persist, with a high number of individuals, in close proximity to wind turbines. However the results show that outside of the core roosting areas the bats prefer foraging habitats further away from the turbines, furthermore they tend to keep further away from the turbines with increasing rotor blade rotation at high wind speeds. This implies a disturbance effect of the turbines, which has also been shown in other studies [12–15], but does not lead to a displacement of the studied Bechstein's bats.

Both wind parks were planned outside the core roosting areas of the bats and did not destroy high quality forest habitat containing high numbers of potential roosts. Additionally, in both wind parks the turbines are operated with curtailment algorithms to prevent collisions. The curtailments also mitigate noise disturbances at night, allowing bats to benefit from the advantages of high quality habitat with roost trees close to the turbines, as the negative disturbance effects are reduced due to curtailments. These examples show, that with careful site planning that excludes sensitive forest habitats, in combination with curtailments in summer, the negative effects of wind turbines on maternity colonies of Bechstein´s bats and probably other narrow-space foraging forest bat species could be mitigated or avoided.

## Supporting information

**S1 File. Table S1 Supporting information on the tagged animals: ID, area, reproductive state, tracking period, number of tracking nights, bearings and valid positions. Table S2** Supporting information on the roosts of the Bechstein´s bats: ID, species, condition, roost height, distance to next turbine, x and y coordinates **Table S3 S3a -S3d:** Roost use of the tagged animals and results of emergence counts. **Table S4:** Size of individual MCPs and number and size of 95%-LoCoH and 50%-LoCoH 50% for all tagged animals.
(DOCX)

**S2 File. html-file including the R-code, results in detail and further comments on the methods used.**
(HTML)

**S3 File. rmd-file including the complete R-code.**
(RMD)

**S4 File. zip-file including data files containing the radio fixes, the habitat mapping and the processed data used for the models.**
(ZIP)

## Acknowledgments

We thank all field workers, who helped in collecting the radio tracking data and the wind park operators, for providing the data from the turbines.

## Author contributions

**Conceptualization:** Johanna Hurst, Robert Brinkmann.

**Data curation:** Johanna Hurst.

**Formal analysis:** Johanna Hurst, Fränzi Korner-Nievergelt.

**Funding acquisition:** Johanna Hurst, Robert Brinkmann.

**Investigation:** Johanna Hurst.

**Methodology:** Johanna Hurst, Fränzi Korner-Nievergelt, Robert Brinkmann.

**Project administration:** Johanna Hurst, Robert Brinkmann.

**Resources:** Johanna Hurst, Robert Brinkmann.

**Supervision:** Johanna Hurst, Robert Brinkmann.

**Validation:** Johanna Hurst, Robert Brinkmann.

**Visualization:** Johanna Hurst, Fränzi Korner-Nievergelt.

**Writing – original draft:** Johanna Hurst.

**Writing – review & editing:** Johanna Hurst, Fränzi Korner-Nievergelt, Robert Brinkmann.

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
