## [Decision Letter · Decision Letter 0]

14 Sep 2025

Dear Dr. Hurst,

We look forward to receiving your revised manuscript.

Kind regards,

Lyi Mingyang, Ph.D.

Academic Editor

PLOS ONE

Journal Requirements:

“Two of the authors (J.H, R.B.) work for the Freiburg Institute of Applied Animal Ecology, which is a consultancy agency, that also conducts wind farm impact

assessment studies. Wind farm operators provided the turbine data for the data analysis. However the wind farm operators had no influence at all on study design, data collection, data analysis, interpretation and publication writing and editing. The authors take full responsibility for the integrity of the study.”

We note that one or more of the authors are employed by a commercial company: Freiburg Institute of Applied Animal Ecology

3. We note that Figures 1-3 & 5-6 in your submission contain [map/satellite] images which may be copyrighted. All PLOS content is published under the Creative Commons Attribution License (CC BY 4.0), which means that the manuscript, images, and Supporting Information files will be freely available online, and any third party is permitted to access, download, copy, distribute, and use these materials in any way, even commercially, with proper attribution. For these reasons, we cannot publish previously copyrighted maps or satellite images created using proprietary data, such as Google software (Google Maps, Street View, and Earth). For more information, see our copyright guidelines: http://journals.plos.org/plosone/s/licenses-and-copyright.

a. You may seek permission from the original copyright holder of Figures 1-3 & 5-6 to publish the content specifically under the CC BY 4.0 license.

Reviewers' comments:

Reviewer's Responses to Questions

**Comments to the Author**

1. Is the manuscript technically sound, and do the data support the conclusions?

Reviewer #1: Yes

Reviewer #2: Yes

Reviewer #3: Yes

2. Has the statistical analysis been performed appropriately and rigorously?

Reviewer #1: Yes

Reviewer #2: Yes

Reviewer #3: Yes

3. Have the authors made all data underlying the findings in their manuscript fully available?

Reviewer #1: Yes

Reviewer #2: Yes

Reviewer #3: Yes

4. Is the manuscript presented in an intelligible fashion and written in standard English?

Reviewer #1: Yes

Reviewer #2: No

Reviewer #3: Yes

Reviewer #1: The manuscript presents VHF radiotelemetry data from 31 female Bechstein’s bats across two forest wind-parks and uses LoCoH home ranges + conditional logistic models to test whether bats avoid turbine proximity and whether turbine operation (rotor speed / wind) affects distance-to-turbine. It is an interesting, policy-relevant study; however important methodological details, statistical choices, and interpretation require clarification and, in some cases, re-analysis to support the claims.

Introduction

Lines 28 – 36:

To add a short paragraph explicitly distinguishing mortality (strike risk) vs avoidance/behavioral disturbance as separate impacts, and why avoidance is ecologically important (habitat loss, functional landscape fragmentation). This helps readers understand the link from acoustic activity declines to colony-level population effects.

Lines 33 – 34:

I think that you should add these two important references as examples to support your sentence: “Wind turbines can have severe negative impacts on wildlife”. I would like to suggest:

Smeraldo, S., Bosso, L., Fraissinet, M., Bordignon, L., Brunelli, M., Ancillotto, L., & Russo, D. (2020). Modelling risks posed by wind turbines and power lines to soaring birds: The black stork (Ciconia nigra) in Italy as a case study. Biodiversity and Conservation, 29(6), 1959-1976.

Estellés‐Domingo, I., & López‐López, P. (2025). Effects of wind farms on raptors: A systematic review of the current knowledge and the potential solutions to mitigate negative impacts. Animal Conservation, 28(3), 334-352.

Lines 41 – 47:

To discuss briefly the spatial and sampling biases inherent to acoustics (detector detection radius, weather dependence) and that some acoustic studies used paired automatic recorders at multiple distances (give example refs) — this makes the contrast to telemetry more explicit.

Lines 49 – 69:

To add an explicit statement of hypotheses (e.g., H1: colonies avoid areas near operating turbines; H2: avoidance increases with rotor rotation/wind speed; H3: high-quality habitat will mitigate avoidance) — this will sharpen the later tests.

Materials & Methods

Lines 72 – 94:

Good to include pre-construction colony knowledge — strengthens inference about persistence. But the temporal mismatch (HR turbines installed ~2004–2006; PL installed 2019) creates heterogeneity in exposure history. Authors should explicitly describe how they account for differences in turbine age and earlier pre-construction baseline surveys when interpreting long-term effects.

The description of curtailment is brief — authors must fully report curtailment thresholds and implementation periods (dates/hours) because curtailment both affects collision risk and could change noise/operation patterns that the study tests. If operators provided curtailment rules, report exact thresholds and any temporal changes. If not available, state that and discuss implications.

Lines 92 – 122:

I think that 31 individuals is reasonable for VHF work on a small forest bat. But each bat tracked only 2–4 nights (short individual monitoring periods). This limits inference about within-individual behavioral variability and temporal response to turbine operation. Compare: many VHF studies of forest bats track individuals for longer periods (e.g., week(s) of continuous tracking) or complement with acoustic sampling (see references cited in manuscript). Recommend authors justify why 2–4 nights gives an unbiased sample of foraging choices (e.g., argument about territoriality and repeated use of core patches).

Cross-bearing VHF is subject to angular error — excluding angles <30° / >150° is standard, but authors should quantify expected locational error (e.g., median error from calibration tests or from simultaneous ground-truthing). If no calibration, estimate error using geometry and observer positions; report a confidence radius for each fix or a mean error. Many telemetry studies report mean triangulation error (e.g., 20–100 m depending on forest). Not accounting for variable locational error can bias LoCoH area estimation and distance-to-turbine measures.

Turbine data were provided at 10-minute resolution. Bearing intervals were 3 min. Authors should explain how they matched fix timestamps to the 10-min turbine dataset (nearest 10-min window? interpolation?). Precise matching procedure affects the wind/rotor analysis.

Authors reduced autocorrelation by using every third fix in the RSF-style analysis — this is pragmatic but coarse. Alternative modern approaches (see below) can model movement and selection jointly (iSSF) or use mixed-effects RSF with correlated random effects. Compare with integrated step selection functions (Thurfjell et al., Avgar et al.) which explicitly account for movement autocorrelation and availability along steps; this is more appropriate for fine temporal telemetry data. Recommend either: (i) re-analyse using iSSF (if step data possible) or (ii) quantify residual autocorrelation after subsampling (e.g., variogram), and show sensitivity to subsampling choice.

Line 131: To all the R codes used in this study in the supplementary materials.

Lines 123–137:

Mapping design seems thorough. Key point: presence points (fixes) were assigned habitat variables from the next mapping point in the same vegetation unit. This assignment method can introduce misclassification if mapping resolution is coarse relative to fix error. Recommend: (i) explicitly state mean distance from each telemetry fix to the assigned mapping point; (ii) test sensitivity by using the nearest mapping point vs. within-unit average; (iii) include mapping error/account for misclassification in the methods or discussion.

LoCoH is a valid choice to respect sharp habitat boundaries (Getz et al. 2007), and they cite that. Good. However, LoCoH performance can be sensitive to the ‘a’ parameter; choosing a = largest distance between two positions may lead to overly large radii for widely dispersed individuals. Authors should show sensitivity analysis of LoCoH size to ‘a’ parameter (e.g., alternative a values, cross-validation). Alternative methods: kernel density estimation (KDE) with bandwidth selection, biased random bridges (for movement autocorrelation), or AKDE (Autocorrelated KDE; using ctmm) which explicitly accounts for temporal autocorrelation and yields confidence intervals for home-range area. Suggest discussing these alternatives and justifying LoCoH over e.g., AKDE or BRB.

Lines 149 – 150:

I think that you should add these two important references as examples to support your sentence: “The roosting area and the home range of the colony and the individals was calculated as minimum convex polygon (MCP) of all radio fixes of the colony resp.”. I would like to suggest:

Ancillotto, L., Palmieri, A., Canfora, C., Nastasi, C., Bosso, L., & Russo, D. (2022). Spatial responses of long-eared bats Plecotus auritus to forestry practices: implications for forest management in protected areas. Forest Ecology and Management, 506, 119959.

Kurek, K., Gewartowska, O., Tołkacz, K., Jędrzejewska, B., & Mysłajek, R. W. (2020). Home range size, habitat selection and roost use by the whiskered bat (Myotis mystacinus) in human-dominated montane landscapes. PloS one, 15(10), e0237243.

Lines 159 – 197:

Using conditional logistic regression (clogit) on matched pairs (use/available) is a standard case-control RSF approach. That is fine. However: the choice of pseudo-absence locations drawn from mapping points within 95% LoCoH — while reasonable — should be justified: mapping points are not uniformly distributed and represent sampling locations, not necessarily unbiased availability. Better would be to sample availability uniformly across the 95% LoCoH polygon (with or without habitat-stratified sampling) or to weight pseudo-absences by mapping effort. The current design risks selection bias if mapping density varies.

Creating 10 datasets to assess random pairing is ad-hoc. A more statistically coherent approach is to use multiple imputation or to sample a large set of random available points per used fix and fit a hierarchical model that integrates uncertainty (or fit a conditional logistic with many available per used). Combining posterior draws from separate dataset fits (i.e., uniting the 4,000 draws from each of 10 datasets) is questionable statistically because it treats the datasets as independent replicates when they are not (they are different random draws from the same underlying availability). A hierarchical Bayesian model that treats availability sampling as part of the model (or fits a single model with many available points per used) is preferable. If authors keep the 10-dataset approach, they must justify pooling procedure formally (e.g., show results are stable across datasets and present median estimates and variability across datasets rather than literally merging posteriors).

Authors must demonstrate that this subsampling adequately reduces temporal autocorrelation (e.g., show semivariogram or autocorrelation function pre/post subsampling). Many modern studies use mixed models with autocorrelation structures or iSSF to directly model movement and selection. Recommend at least a sensitivity check (every 2nd, 4th fixes) and reporting how estimates change.

removed correlated variables (r>0.8 for leaf/herb). Good, but should present a correlation matrix (Supplement) and Variance Inflation Factors (VIFs) for final variables.

Authors state individual ID was random effect — good. But conditional logistic with random intercepts in stan_clogit must be carefully parameterized; clarify how random effects were implemented (by stratum or as intercepts) and report between-individual variance.

They report checking R̂ and effective sample size. Good; include R̂ values and n_eff in Supplement. Also report posterior predictive checks, and ideally model comparison (e.g., WAIC / LOO-CV) for models with/without turbine interaction terms.

They included interactions between distance to turbine and habitat variables. This is important. But interactions can be nonlinear — consider smoothing (GAM) or categorize distance bands (e.g., 0–100 m, 100–300 m, >300 m) to test nonlinearity or check the linearity assumption on logit scale.

Lines 197 – 205:

Distance to turbine is nonnegative, skewed, and bounded by study geometry. Authors must show residual diagnostics (they say visually assessed) and consider transformations (log distance) or GLMM families if residuals deviate. Also spatial nonindependence (fixes clustered in space) may violate assumptions; consider spatial random effects or include spatially structured error (e.g., via INLA/GAM).

The effect of rotor speed/wind on avoidance may be nonlinear with thresholds (e.g., turbines start above a cut-in speed). Authors hint at thresholds (different in HR and PL). Consider testing for breakpoints (piecewise regression) or including turbinerunning (on/off) as a categorical variable. Report the cut-in speeds and rotor RPM thresholds explicitly.

They assigned turbine metrics of the “next turbine” to each fix. Clarify how “next turbine” is defined (closest by Euclidean distance?); if a bat is between turbines, its behavior may respond to multiple turbines simultaneously. Consider aggregating metrics across nearest k turbines (e.g., nearest 3) weighted by inverse distance or modeling turbine presence as a spatial covariate (distance to nearest turbine + whether nearest turbine is running).

Curtailment means turbines may be turned off at low wind speeds (and bat activity may be highest at those low speeds). This confounds wind speed, rotor rotation and bat distance. Authors should present turbine operational state (running vs stopped) as separate predictor and test its effect.

Results

Lines 206 – 240:

Manuscript: HR: 14 roost trees (6 in 2019, 9 in 2022); roost distances to turbines mean 415 ± 225 m (range 150–811 m). PL: 26 roost trees; roost-turbine distances mean 295 ± 161 m (range 70–684 m). Colony sizes reported from emergence counts (max 88 in HR, 59 in PL).

Present roost coordinates or a table (S1) with roost IDs, dates, and nearest turbine distance — this helps reproducibility. If privacy concerns exist, provide masked coordinates but with exact distances to turbines.

The narrative on roost movement (HR core moving north compared to 2007) should be supported with statistical comparison (e.g., difference in centroid distance to turbines between years) rather than visual description. Provide metrics with confidence intervals.

Lines 241 – 259:

Manuscript: Colony MCPs sizes, distribution of fixes relative to roost (percent within 500 m and 1,000 m), LoCoH 95% and 50% ranges, core areas.

To provide tables with individual MCP and LoCoH areas (they mention Table S3 — ensure it is comprehensive). Show variability across individuals and years.

The authors report large variability in LoCoH sizes — include a sensitivity test showing how LoCoH a-parameter choice affects results.

The test for territoriality (core areas overlap) is descriptive; consider quantifying overlap (Bhattacharyya index, utilization distribution overlap index) and test whether overlap is less than expected by chance.

Lines 267 – 277:

Bats used habitats close to turbines mainly when close to their roosts and when high DBH and low shrub/herb cover (high quality). The distance to turbines increased with rotor rotation at high wind speeds; different thresholds between HR (8 m/s) and PL (12 m/s). Figures present model outputs (Fig 7, Fig 8).

Present the model coefficients (posterior means and 95% CIs) in a table for the final model (they refer to S1 Table S4 — ensure this is included). For interaction terms, present marginal effect plots (they have Fig 7 — good) but also show intervals across individuals, because heterogeneity could be large.

On pooling posterior draws from 10 datasets: instead of pooling, present parameter estimates per dataset to show stability, and then report combined estimates using formal model averaging or hierarchical pooling. The current description (“the 4,000 simulated values from each of the 10 data sets were united and used to describe the averaged effects”) is ambiguous and statistically weak; explain procedure and justify.

For the wind/rotor model: show predicted distances for representative roost distances and study areas; show whether effect sizes are biologically meaningful (e.g., how many meters change per 1 m/s). Report effect magnitudes with credible intervals so readers can judge ecological significance.

Discussion

Lines 291 – 321:

Manuscript conclusion: colonies did not abandon roosts near turbines; colony sizes remain high. This is a central and important finding.

Persistence does not equal absence of negative effects. The authors acknowledge this, but must discuss alternative possibilities: demographic compensation (e.g., immigration), delayed population declines, or reduced fitness (lower juvenile survival) not detected by short tracking windows. Recommend tempering the language and calling for long-term demographic monitoring. Also discuss that the colonies may tolerate turbines up to a threshold of turbine density/operation.

Lines 337 – 350:

Bats prefer high-quality habitat close to roosts despite turbines; further from roosts they avoid turbine-proximate foraging. This aligns with acoustic studies showing activity declines near turbines. Good linkage.

To discuss how the limited tracking nights and possible seasonal effects (they tracked only late June–July) may bias detection of avoidance (e.g., seasonal changes in foraging range/juvenile mobility). Recommend recommending multi-season tracking.

Lines 351–360:

The authors infer that increased rotor rotation (and assumed noise) leads to increased distance; this is plausible and supported by other studies (they cite Ellerbrok et al.).

Rotor rotation may also correlate with other cues (vibration, infrasound, blade movement airflows). To support noise mechanism more directly, authors could (in future) include in-forest sound pressure level (SPL) measurements at different distances and turbine operational states — this would strengthen causal inference. At a minimum, the discussion should explicitly note that rotor RPM is a proxy for noise, not a direct measurement.

Reviewer #2: In this manuscript, the authors investigate how wind turbines built in forests influence the roosting and foraging behaviour of two Bechstein’s bat maternity colonies in Germany. Using radio telemetry, they determined the locations of day roosts during summer and identified the areas used at night by tagged individuals. These locations were then analysed in relation to several habitat parameters, as well as to distances from roosts and wind turbines, in order to assess the effects of the latter.

Given the increasing pressure to identify suitable sites for wind farms and the resulting trend to place turbines within forests, this study addresses an important and timely question in the conservation ecology of forest-dwelling bats. The telemetry data collected from these maternity colonies near wind turbines provide a rare opportunity to investigate the effects of forest wind farms on both individual behaviour and colony ecology, going beyond the commonly used approach of acoustic monitoring of bat activity. I think the results are quite interesting and the analysis appears to be well executed and robust. However, to improve readability and clarify the presentation of methods and results, the manuscript would benefit from substantial language revision.

I have a few substantive comments, as well as several minor, mostly linguistic points that should be addressed prior to publication. Below (in the attached document), I outline these points by manuscript section, referring to the line numbers of the manuscript. I have highlighted certain passages as examples that, in my view, could benefit from language improvement, and in some cases I offer possible suggestions for revision. These examples and suggestions are intended for guidance and are not intended to be prescriptive or exhaustive. For example, I have not identified every instance where a comma is needed.

Reviewer #3: General Comment

This is an interesting study that clearly involved a great deal of field effort, and the data and analysis you present have the potential to make a valuable contribution. However, the manuscript would greatly benefit from a thorough language revision, including attention to sentence flow and punctuation, to improve clarity and readability. In addition, the description of the methods should be made clearer and more transparent so that readers can fully understand and assess your approach. I also noticed that the results of the Bayesian models are not fully presented. It would strengthen the manuscript if you included the main results (e.g., estimates, credible intervals, etc.) in the main text, ideally in a table or figure, and provided the full model outputs in the supplementary material.

Line-specific comments

• Lines 34–35: Besides direct collisions, bats may also die due to barotrauma associated with wind turbines.

• Lines 48–49: I think the impact of wind turbines on forest specialists is not only (or primarily) linked to noise disturbance, but also to habitat degradation and fragmentation. Moreover, forest specialists are already heavily impacted by habitat loss from many other anthropogenic pressures unrelated to wind parks, and wind parks add to these cumulative effects.

• Line 53 (and throughout): The subject of sentences sometimes changes in a way that is misleading or unclear. For example: “As maternity roosts Bechstein’s bats use nearly exclusively tree roosts (22–25). Like other forest bats it switches the roosts often…” The subject shifts here in a way that disrupts clarity. I recommend reviewing these passages for consistency.

• Line 105: I would avoid stating that the weight of the transmitter is “harmless” for bats. The cited publication is relatively old, and a recent review (https://onlinelibrary.wiley.com/doi/full/10.1111/mam.12369) highlights that there is a lack of studies explicitly testing the effect of tag weight, and that tags in general can affect bat behaviour and body condition. While 5% is indeed a conservative and commonly used threshold, it should not be presented as harmless.

• Line 110: The description of how fixes were categorized could be clearer. For example, were all night fixes assumed to represent foraging? Lactating females often return to the roost during the night to nurse their offspring, so some night fixes might actually correspond to nursing rather than foraging.

• Line 124: Abbreviations should be defined at first use. In this case, MCP is only explained later in the text.

• Lines 129–132: This sentence is unclear. Were the points chosen randomly using an algorithm in R? Or is R mentioned here simply because it was the software used? Since you already specify later that all analyses were conducted in R, I would suggest avoiding mention here unless there is a particular methodological reason.

• Line 136: Could you clarify “10 m around what point exactly”? It seems like some detail may be missing here.

• Lines 147–148: How did you determine whether bats were inside the roosts? Were you always able to locate the roost itself? This would also help clarify the concern raised at line 110.

• Line 148: If this is the dataset after excluding the positions mentioned above, I would suggest referring to it as the “final dataset” rather than the “total dataset.”

• Lines 168–172: I am not entirely sure I follow this part. From what I understand, the 10 datasets contained the same fixes paired with different, randomly generated pseudo-fixes. Is that correct? Or did you randomly split the fixes and pseudo-fixes into 10 subsets? I think this section would benefit from a clearer explanation.

• Lines 193–194: I am unclear about why the draws from different models were merged. Could you provide some justification or explanation for this step?

• Lines 210 and 223: The maximum distances between roosts reported seem quite high for Bechstein’s bats. I would suggest making it clearer early on (as you do later in the discussion) that you cannot distinguish between different colonies or stable subgroups within colonies.

• Line 315 (and elsewhere): Please be cautious when stating that colony sizes were so high (and similar colony-specific statements later). As you note earlier in the discussion, you cannot conclusively determine whether these were single colonies.

• Lines 363–364: You may want to add some discussion on the site fidelity of Bechstein’s bats. Your results suggest the colonies remain in the area, but Bechstein’s bats are known for strong fidelity to roosting areas and may continue using them due to their relative lack of flexibility. Other forest specialists might respond differently, and this contrast could be worth noting.

**Do you want your identity to be public for this peer review?** For information about this choice, including consent withdrawal, please see our Privacy Policy

Reviewer #1: No

Reviewer #2: No

Reviewer #3: No

---

## [Author Response · Author response to Decision Letter 1]

2 Dec 2025

We thank the editor and all 3 reviewers for the helpful comments, which helped to improve our manuscript remarkably.

All comments are responded in detail in the file "Response to reviewers".

---

## [Decision Letter · Decision Letter 1]

6 Jan 2026

Dear Dr. Hurst,

plosone@plos.org . A letter that responds to each point raised by the academic editor and reviewer(s). You should upload this letter as a separate file labeled 'Response to Reviewers'.A marked-up copy of your manuscript that highlights changes made to the original version. You should upload this as a separate file labeled 'Revised Manuscript with Track Changes'.An unmarked version of your revised paper without tracked changes. You should upload this as a separate file labeled 'Manuscript'.

We look forward to receiving your revised manuscript.

Kind regards,

Lyi Mingyang, Ph.D.

Academic Editor

PLOS One

Journal Requirements:

Reviewers' comments:

Reviewer's Responses to Questions

**Comments to the Author**

Reviewer #2: All comments have been addressed

2. Is the manuscript technically sound, and do the data support the conclusions?

Reviewer #2: Yes

3. Has the statistical analysis been performed appropriately and rigorously?

Reviewer #2: Yes

4. Have the authors made all data underlying the findings in their manuscript fully available?

Reviewer #2: Yes

5. Is the manuscript presented in an intelligible fashion and written in standard English?

Reviewer #2: Yes

Reviewer #2: The authors have done a good job revising the manuscript in response to the reviewers’ comments. In particular, the language has improved significantly, resulting in much better readability, flow, and clarity in the revised version. In their responses to the reviewers, I would have appreciated more consistent provision of line numbers referring to the revised manuscript, as this would have made it easier to cross-check the implemented changes.

I have a few additional minor comments and questions; otherwise, I believe the manuscript is close to being ready for publication. The line numbers provided below refer to the clean manuscript without tracked changes.

Please find my comments and questions in the attached Word document.

**Do you want your identity to be public for this peer review?** For information about this choice, including consent withdrawal, please see our Privacy Policy

Reviewer #2: No

---

## [Author Response · Author response to Decision Letter 2]

19 Jan 2026

All responses can be found in the uploaded document "response to reviewers.docx"

---

## [Decision Letter · Decision Letter 2]

24 Feb 2026

Habitat use of Bechstein´s bats (Myotis bechsteinii) in wind parks in forests

PONE-D-25-42548R2

Dear Dr. Hurst,

We’re pleased to inform you that your manuscript has been judged scientifically suitable for publication and will be formally accepted for publication once it meets all outstanding technical requirements.

Kind regards,

Lyi Mingyang, Ph.D.

Academic Editor

PLOS One

Additional Editor Comments (optional):

Well done!

Reviewers' comments:

Reviewer's Responses to Questions

**Comments to the Author**

Reviewer #2: All comments have been addressed

2. Is the manuscript technically sound, and do the data support the conclusions?

Reviewer #2: Yes

3. Has the statistical analysis been performed appropriately and rigorously?

Reviewer #2: Yes

4. Have the authors made all data underlying the findings in their manuscript fully available?

Reviewer #2: Yes

5. Is the manuscript presented in an intelligible fashion and written in standard English?

Reviewer #2: Yes

Reviewer #2: (No Response)

**Do you want your identity to be public for this peer review?** For information about this choice, including consent withdrawal, please see our Privacy Policy

Reviewer #2: No

---

## [Editor Report · Acceptance letter]

PONE-D-25-42548R2

PLOS One

Dear Dr. Hurst,

I'm pleased to inform you that your manuscript has been deemed suitable for publication in PLOS One. Congratulations! Your manuscript is now being handed over to our production team.

Kind regards,

on behalf of

Professor Lyi Mingyang

Academic Editor

PLOS One